# Multifunction Web-like Polymeric Network Bacterial Cellulose Derived from SCOBY as Both Electrodes and Electrolytes for Pliable and Low-Cost Supercapacitor

**DOI:** 10.3390/polym14153196

**Published:** 2022-08-05

**Authors:** Muhamad Hafiz Hamsan, Norhana Abdul Halim, Siti Zulaikha Ngah Demon, Nurul Syahirah Nasuha Sa’aya, Mohd Fakhrul Zamani Kadir, Zul Hazrin Zainal Abidin, Nursaadah Ahmad Poad, Nurul Farhana Abu Kasim, Nur Amira Mamat Razali, Shujahadeen B. Aziz, Khairol Amali Ahmad, Azizi Miskon, Norazman Mohamad Nor

**Affiliations:** 1Department of Physics, Centre for Defence Foundation Studies, National Defence University of Malaysia, Sungai Besi Camp, Kuala Lumpur 57000, Malaysia; 2Centre for Tropicalization, National Defence University of Malaysia, Sungai Besi Camp, Sungai Besi, Kuala Lumpur 57000, Malaysia; 3Faculty of Defence Science & Technology, National Defence University Of Malaysia, Sg Besi Camp, Sungai Besi, Kuala Lumpur 57000, Malaysia; 4Physics Department, Faculty of Science, Universiti Malaya, Kuala Lumpur 50603, Malaysia; 5Centre for Ionics University of Malaya (C.I.U.M.), Department of Physics, Faculty of Science, Universiti Malaya, Kuala Lumpur 50603, Malaysia; 6Hameed Majid Advanced Polymeric Materials Research Lab., Physics Department, College of Science, University of Sulaimani, Qlyasan Street, Kurdistan Regional Government, Sulaimani 46001, Iraq; 7The Development Center for Research and Training (DCRT), University of Human Development, Kurdistan Region of Iraq, Sulaymaniyah 46001, Iraq; 8Faculty of Engineering, National Defence University of Malaysia, Kem Sg Besi, Kuala Lumpur 57000, Malaysia

**Keywords:** EDLC, supercapacitor, bacterial cellulose, biopolymer, green energy, electrolyte

## Abstract

In this work, bacterial cellulose (BC)-based polymer derived from a symbiotic culture of bacteria and yeast (SCOBY) are optimized as both electrodes and electrolytes to fabricate a flexible and free-standing supercapacitor. BC is a multifunction and versatile polymer. Montmorillonite (MMT) and sodium bromide (NaBr) are used to improve mechanical strength and as the ionic source, respectively. From XRD analysis, it is found that the addition of MMT and NaBr has reduced the crystallinity of the electrolyte. Most interaction within the electrolyte happens in the region of the OH band, as verified using FTIR analysis. A maximum room temperature conductivity of (1.09 ± 0.02) × 10^−3^ S/cm is achieved with 30 wt.% NaBr. The highest conducting SCOBY-based electrolytes have a decompose voltage and ionic transference number of 1.48 V and 0.97, respectively. The multiwalled carbon nanotube is employed as the active material held by the fibrous network of BC. Cyclic voltammetry shows a rectangular shape CV plot with the absence of a redox peak. The supercapacitor is charged and discharged in a zig-zag-shaped Perspex plate for 1000 cycles with a decent performance.

## 1. Introduction

Cellulose is a natural and abundant polymer with excellent mechanical properties [1]. Various sources of cellulose, e.g., plant, animal, and bacteria, have been employed in different industries such as cosmetic, pharmaceutical, textile, bioadhesive, and mucoadhesive drug delivery systems [2]. Among all kinds of cellulose, bacterial cellulose (BC) has gained considerable attention in electrolyte application due to its unique characteristics, for instance, high aspect ratio, formability, eco-friendliness, excellent mechanical strength, flexibility, and ability to act as an ion conductor [3]. BC has ether groups (–C–O–C–) in its structure, which is almost similar to polyethylene oxide (PEO). Glucose monomers are connected via –C–O–C– functional groups in the nanofiber structure of BC [4]. This kind of structure is beneficial for dissolving salts. Each oxygen in the structure of BC has electron lone pairs, which will form a dative bond with salt [5]. The properties of BC can be modified by polymer blending. The efficiency of polymer synthesis typically depends on the method of culturing. In static culture, it is crucial to monitor the pH level. The development of acids as products of the metabolism of microorganisms can decrease the effectiveness of synthesis [6].

Li et al. [7] reported that blending BC with PEO has improved the thermal stability and mechanical strength of the film. The formation of dendrites has to be the main problem in energy devices study, especially for lithium batteries. The authors also stated that the BC presence in electrolytes could eliminate the growth of dendrites on the surface of the electrodes. According to Yue et al. [8], the gel polymer electrolyte system of sulfonated BC-polyaniline has achieved 5.2 × 10^−3^ S cm^−1^. The authors claimed that the decomposition of polyaniline film increased when it was reinforced with BC. The preparation of BC plays a vital role in achieving high-conducting polymer electrolytes. Sabrina et al. [9] concluded that polymer electrolyte from freeze-dried BC has a larger porous structure and ionic conductivity than oven-dried BC. Luo et al. [10] reported electrical double-layer capacitors (EDLC) comprising BC, acetylene black, and activated carbon as the electrode. The author claimed that BC could form a free-standing film that can avoid using the current collector, reducing the overall weight and price.

The presence of BC gives hope in fabricating the current demand for abnormal-shaped energy storage devices. EDLCs are one of the capable energy storage devices that can provide high power density with fast charge–discharge cycles while being pliable. Ions undergo an adsorption/desorption process on the surface of carbon electrodes; thus, bending the EDLC might affect their performance [11]. The unique characteristics of EDLCs are a long life cycle, rapid charging/discharging process, high power density, and a straightforward and safe fabrication method [12]. In EDLC devices, many types of carbon have been incorporated as electrode materials, for instance, graphite [13], aerogel [14], carbon nanofibers [15], carbon nanotube [16], and activated carbon [17]. Carbon nanotube (CNT) has been extensively studied among these carbons due to their excellent electron mobility and mechanical properties. Typically, CNT will form a porous network that is beneficial for the movement of ions [18]. CNT can form a hydrogen bond or van der Waals force with polymer, thus providing flexible film. Hence, CNT is a vital material to fulfill the needs of modern society.

This zig-zag-shaped perspex plate is employed to test the supercapacitor’s performance with irregular shapes instead of flat shapes. This design enables the SCOBY-based supercapacitor to be useful in electronic applications that require irregular shapes, such as curved, wearable, miniaturized, portable, and flexible consumer electronics. Furthermore, the materials and methods are abundant, inexpensive, and straightforward. Therefore, large-scale production of this SCOBY-based supercapacitor is possible in the future. In this work, BC, NaBr, and MMT are used as the polymer host, ion provider, and filler, respectively. BC is also used in the electrodes as the polymer matrix with hydroxyl multiwalled CNT (MWCNT-OH) as the active material.

## 2. Materials and Methods

### 2.1. Materials

Carbon nanotubes and sodium dodecyl sulfate were obtained from Sigma–Aldrich (Saint Louis, MO, USA). The source of cellulose is a type of bacterial cellulose (BC) hydrogel. BC will degrade naturally in pH 5 and must be filtered and washed with deionized water before being used. The weight-average degree of polymerization (DPw) of bacterial cellulose remained in the range of 14,000 of 16,000 during cultivation at pH 4, but at pH 5, the DPw decreased from 16,800 to 11,000. Figure 1 shows the fresh BC and degraded BC. Sodium bromide was purchased from SYSTERM.

### 2.2. Preparation of BC-NaBr-MMT (BXD) Electrolytes

For bacteria cellulose-based electrolytes (BXD), 0.05 g montmorillonite and 3.0 g bacterial cellulose were sonicated for 30 min in 50 mL of deionized water. Next, the sample was stirred using magnetic stirring for 24 h (60 °C/700rpm). The next step was Buchner funnel filtration, and the BXD sample was left to dry for 30 min before the inclusion of 0 to 50 wt.% sodium bromide. The final sample was left to dry for 24 h at room temperature. Electrolyte with 0, 10, 20, 30, 40, and 50 wt.% NaBr was named as BXD, BXD10%, BXD20%, BXD30%, BXD40%, and BXD50%, respectively.

### 2.3. Characterization of BC-NaBr-MMT Electrolytes (BXD)

HIOKI 3532–50 LCR impedance spectroscopy was set from 50 Hz to 50 MHz to study the effect on frequency of the ionic conductivity (*σ*). The BXD was pressed in between two stainless steel surfaces of a Teflon case. The *σ* was obtained via the following equation:*σ* = *t*/(*AR_B_*)(1)

Here, *R_B_* is the bulk resistance of the electrolyte. The thickness (*t*) of BXD film was measured and maintained using a Mitutoyo micrometer screw gauge. The electrochemical stability window of BXD films was analyzed using linear sweep voltammetry (LSV) analysis. In this part, the potential was swept linearly from 0 to 3 V. This analysis was conducted using a Digi-IVY DY2300 potentiostat at a slow scan rate of 10 mV/s. The dominancy of ions as an ionic conductor was confirmed using transference number analysis (TNM). The ionic (*t_ion_*) and electronic (*t_elec_*) and transference number (TNM) were analyzed using the V&A instrument DP3003 digital DC power supply, which is in series with a multimeter for current monitoring. The working voltage for the polarization process was 0.2 V. The analysis was performed at room temperature where *t_ion_* can be obtained via the given equation:*t_ion_* = (*I_i_*−*I_s_*)/*I_i_*(2)
where stabilized and initial currents are denoted as *I_s_* and *I_i_*, respectively. In order to verify the crystallinity of BC and CNT, X-ray diffraction (XRD) analysis was conducted using Bruker Model X-Ray Diffractometer Bruker Model. The range in this study was from 10° to 100° at a step size of 0.002°. X-ray of 1.5406 Å wavelengths were generated by a Cu Kα source. The selected XRD data were then deconvoluted using Origin 9.0 software via Gaussian function to determine the degree of crystallinity (*χ_c_*). *χ_c_* can be calculated using the following equation:*Χ_c_* = (*A_c_/A_t_*) × 100%(3)

Here, *A_c_* and *A_t_* are area of crystalline region and total area, respectively.

### 2.4. Preparation of BC-CNT Electrodes (BXC)

BC (3.0 g) was filtered using manual filtration. An amount of 0.05 mg of hydroxyl (MWCNT-OH) (conductivity, S = 107 S/m) was dispersed in 20 mL of deionized water to prepare the MWCNT solution. BC and MWCNT-OH were then blended with deionized water using a magnetic stirring process for 24 h at 45 °C until the homogenous solution was obtained. The homogenous BXC solution was then vacuum-filtered using a Buchner funnel. Finally, the prepared BXC nanocomposites electrodes were peeled off and dried at ambient temperature for further characterization. Figure 2 shows the prepared BXD electrolyte and BXC electrodes from bacterial cellulose.

### 2.5. Construction of Full Bacterial Cellulose-Based EDLC

The EDLC was prepared by sandwiching the BXD electrolyte between two identical BXC electrodes. Aluminum was cut into the desired shape and placed on top and bottom of the top and bottom BXC, respectively. The EDLC was then vacuumed, sealed, and packed in nonconductive plastic wraps. The packed EDLC was placed in a zig-zag-shaped perspex plate where part of the aluminum foil stuck out on the side. This part was connected to the Neware battery cycler and potentiostat for charge–discharge analysis and cyclic voltammetry, respectively. The fabricated EDLC is shown in Figure 3.

### 2.6. Characterization of Full Bacterial Cellulose-Based EDLC

Cyclic voltammetry analysis was selected to verify any redox reaction in the potential range used in this work. Digi-IVY DY2300 potentiostat was used for this analysis with many scan rates at 10, 20, 50, and 100 mV/s. Apart from that, CV enables us to identify the specific capacitance (*C_cyc_*) of the fabricated EDLC using the following equation:*C_cyc_* = ∫ *I*(*V*)*dV*/2*mx*(*V_f_* − *V_i_*)(4)
where *x* stands for the scan rate and *m* is the mass of active material, which, in this case, is the mass of CNT. *V_i_* and *V_f_* are the initial and final voltage, respectively. *I*(*V*) is the area of the CV plot using the final and initial voltage as the maximum and minimum value, respectively. The area of the CV plot was determined using OriginPro software. The NEWARE battery cycler with a current density of 0.143 mA/cm^2^ was set to test the charge–discharge properties of the constructed EDLC. Important parameters of the EDLC, for instant specific capacitance (*C_s_*), equivalent series resistance (*ESR*), power density (*P_den_*), and energy density (*E_den_*):*C_s_* = *i*/*gm*(5)
*ESR* = *V_drop_*/*i*(6)
*E_den_* = (*C_s_V*^2^)/2(7)
*P_den_* = *V*^2^/(4*mESR*)(8)
where *g* stands for the gradient of the discharge part, *V_drop_* stands for the potential drop, and *i* is the working current, which is 1 mA.

## 3. Results

### 3.1. BC-NaBr-MMT (BXD) Electrolytes Study

#### 3.1.1. Ionic Conductivity Analysis of the Bacterial Cellulose Electrolytes

A polymer electrolyte must possess a high ionic conductivity just to be an efficient ion conductor in an EDLC. The conductivity must be at least ~10^−3^ S/cm. As shown in Figure 4, mixing BC and MMT produces a low conductive film with a conductivity of (1.24 ± 0.01) × 10^−9^ S/cm. This value lies between an insulator and a conductor. High conductivity is obtained with more amount of NaBr. As salt is added, instead of the cellulose matrix, the matrix is filled with Na^+^ and Br^−^, which will be beneficial for the adsorption or intercalation process in the energy devices [19].

As a consequence, high ionic conductivity is obtained. The maximum conductivity achieved is (1.09 ± 0.02) × 10^−3^ S/cm with 30 wt.% NaBr. However, adding more salt is not always the best way to improve conductivity. This is due to limited complexation sites in the polymer matrix [20]. In a crowded space of polymer matrix, free ions tend to recombine and recrystallized, which block the ionic pathway causing a significant drop in ionic conductivity value. In addition, recrystallized salt can reduce the performance of the EDLC, especially power density. Thus, this explains the drop in ionic conductivity as 40 wt.% and 50 wt.% NaBr is employed in the BC-MMT polymer matrix.

#### 3.1.2. Crystallinity Analysis of the Bacterial Cellulose Electrolytes

In addition to high ionic conductivity, a good biopolymer electrolyte must be amorphous or have a low crystallinity to be useful in energy storage applications. Figure 5 shows the XRD spectrum for BC, MMT, and BXD electrolytes with various weight percentage of NaBr from 2theta of 10° to 100°. It can be seen that MMT has three distinguishable crystalline peaks at 2theta = 17.6°, 26.6°, and 28.5°. These obtained crystalline peaks of MMT are comparable to other reported MMT works [21,22]. The MMT peaks at 2theta = 17.6°, 26.6°, and 28.5° in this work correspond to d-spacing of 5.08 Å, 3.35 Å, and 3.03 Å, respectively [22]. Meanwhile, BC has crystalline peaks at 14.6° with a shoulder peak at 17.0 and at 2theta = 22.9°. The peaks at 14.6° and 17.0° belong to the crystal lattice plane of (11 0) and (110), respectively. These peaks are common native peaks for a cellulose, and thus these outcomes are almost similar with other studies [23,24,25].

When BC is blended with MMT (BXD0%), all crystalline peaks of BC and MMT are suppressed to a lower intensity. This is a proof of interaction between polymer chain with MMT via hydrogen bond. As 30 wt.% NaBr is added, most crystalline peaks become smaller and less sharp, signifying an improvement in the amorphous structure of the electrolyte. As the weight percent of NaBr goes beyond 30 wt.%, sharp crystal peaks can be seen in the XRD spectrum of BXD40% and BXD50%. New peaks that appeared in BXD40% and BXD50% correspond to the typical crystal peaks of NaBr, which are comparable to other NaBr work [26]. This explains the drop in conductivity value as 40 and 50 wt.% are added as portrayed in Figure 4.

The deconvolution method was performed on the selected samples to verify the pattern of crystallinity as MMT and NaBr were added into BC polymer matrix. Figure 6 shows the deconvoluted XRD pattern of BC, BXD0%, and BXD30%. BC in Figure 6a shows several sharp and narrow crystalline peaks along with two broad amorphous peaks. This is a typical structure of semicrystalline materials. The *χ_c_* is tabulated in Table 1. As MMT is added into BC polymer chain, the intensity of crystalline peaks of BC are observed to decrease and become smaller. Several crystalline peaks belong to MMT appeared in the deconvoluted XRD pattern of BXD0% (Figure 6b). The slight drop in degree of crystallinity can be seen in Table 1 as MMT is added. This signifies that the addition of MMT enhanced the amorphousness of BC film. In Figure 6c, the intensity of all crystalline peaks of MMT and BC decrease with the introduction of 30 wt.% NaBr, which indirectly reduces the *χ_c_* and improves amorphousness of the film.

#### 3.1.3. Complexation within the Bacterial Cellulose Electrolytes

Typical polymer or cellulose has functional groups with lone pair electrons. This lone pair electron is where most ions or particles can form complexation via a dative bond [27]. This interaction or complexation can usually be detected through shifting peaks in the FTIR spectrum, as seen in Figure 7. The FTIR spectrum of MMT has several familiar peaks at a wavenumber of 3628 cm^−1^, which belongs to the stretching vibration of silanol groups. In comparison, 3416 cm^−1^ and 1634 cm^−1^ are attributed to the OH-adsorption peaks. Peaks between 650 cm^−1^ to 1320 cm^−1^ are characteristic peaks of MMT. Peaks at 992 cm^−1^ and 792 cm^−1^ indicate the region of AlAlOH and AlMgOH bending vibrations, respectively. The peaks of MMT in this work are comparable to other MMT works [28,29].

Meanwhile, for BC, typical cellulose peaks are located at 3340 cm^−1^, 2900 cm^−1^, and 1630 cm^−1^. These peaks correspond to OH stretching vibration, C-H stretching, and C=O vibration, respectively. Peaks located at 1160–1060 cm^−1^ belong to C-O stretching [30,31]. As BC and MMT are blended, the pattern of the FTIR spectrum appears more similar to BC, with the OH peak at 3340 cm^−1^ owing to the high amount of BC compared to MMT. C=O of BC at 1630 cm^−1^ is now at 1638 cm^−1^ with a slight change in the intensity. This verifies the interaction between BC and MMT via hydrogen bond. By referring to the FTIR peak of BXD30% (highest conductivity), OH peaks shifted to a new wavenumber of 3404 cm^−1^. The addition of salt changed the intensity and location of C=O to 1616 cm^−1^. This indicates that ions from the salt interacted with the BC polymer chain.

#### 3.1.4. Surface Morphology of the Bacterial Cellulose Electrolytes

The surface morphology of BC, MMT, and selected electrolyte with NaBr is shown in Figure 8. BC in Figure 8a shows an entangled structure and random multiweb distribution. There are many spaces and gaps between BC fibers that allow insertion of other compounds into the BC matrix, which enhances the BC properties [32]. The diameter of the BC fiber is between 57 and 192 nm. Jia et al. [33] and Pal et al. [34] reported that the diameter of BC fiber was 71 nm and 100 nm, respectively. Thus, our BC has almost similar size distribution with other BC-based works.

Figure 8b depicts the surface of MMT where it has a sheet structure along with large leaf-like crystals forming a dense aggregate. As MMT is added into BC matrix (Figure 8c), the diameter of BC fibers reduces to 34.76 from 57.95 nm. The gap between BC fibers is filled by MMT particles. Small MMT particles can penetrate into BC matrix, while large MMT particles accumulate at the surface of BC fiber. The existence of OH groups in both BC and MMT can lead to weak organic–inorganic hydrogen bonding interactions. This signifies the interaction between MMT and BC structures. Similar observations are obtained in other MMT-BC-based works [35]. In Figure 8d, particles from NaBr salt are embedded in the structure of BC-MMT matrix. The impregnation of NaBr further reduces the size of the BC fiber to 26.05 from 41.61 nm. Ions from the salts form dative bonds with oxygen containing functional groups in BC and MMT, causing the unraveling of the BC fibers network. The result is comparable with Zhou et al. [36], where a similar phenomenon was obtained as BC was doped with sodium alginate. The outcomes from FESEM analysis are consistent with the trend in XRD and FTIR analyses.

#### 3.1.5. Contribution of Ions and Electrons in The Bacterial Cellulose Electrolytes

It is well known that in an EDLC, ions interact with the surface of carbon electrodes through electrostatic force. In this study, Na^+^ and Br^−^ are the dominant charge-exporting species, and electrons are the less dominant charge exporter. It is crucial to verify the contribution of ions to the overall conductivity. This can be obtained by conducting polarization on the electrolyte with constant working voltage where the changes in current flow are being monitored simultaneously. The TNM plot of BXD30% is illustrated in Figure 9. A high current of 23.7 μA can be seen just when the process started. At this early stage, ions and electrons are driven toward the stainless steel electrodes, thus providing a high current value. In less than 10 s, a drastic drop in the current value can be observed. As soon as ions reach the surface of stainless steel, ions form electrostatic force with the electrodes.

Ions cannot pass through stainless steel, and only electrons can flow through stainless steel. This explains the low current reading of 0.7 μA beyond 10 s. This is where the polarization process starts when we observe a stable and constant current reading. With all this information from TNM analysis, *t_ion_* and *t_elec_* for BXD30% are 0.97 and 0.03, respectively. This value is comparable to other polymer-Na salt-based electrolyte systems. Shetty et al. [37] reported that carboxyl methylcellulose-NaNO_3_ had a *t_ion_* of 0.97. According to Bhargav et al. [38], the presence of sodium salt in PVA provided a series of *t_ion_* between 0.954 and 0.974.

#### 3.1.6. Potential Limit Test for the Bacterial Cellulose Electrolytes

In this modern era, many technology or electrical appliances require a certain amount of voltage. Thus, it is essential to know the potential stability of an electrolyte before the EDLC fabrication process. Using a safe potential range can avoid the breakdown of the polymer chain during the long charging and discharging process. The goal is to identify at which potential the polymer electrolyte begins to oxidize or reduce and, thus, degrade. This is sometimes called breakdown potential [39]. The LSV plot of the highest conducting electrolyte is depicted in Figure 10. A stable current flow can be seen as the potential swept linearly from 0 to 1.48 V. Thus, it is safe to say that stable polarization and charge-double layer can be achieved in this potential region. The slope of the LSV plot is observed to increase more as the potential reaches 2.2 V. Beyond this limit, more degradation happens in the polymer matrix. This can reduce the performance of the EDLC.

### 3.2. Full Bacterial Cellulose-Based EDLC Study

#### 3.2.1. Important Storage Properties

The full SCOBY-based EDLC is successfully fabricated where the charge–discharge curve of it is shown in Figure 11. The shape of the curves is a typical plot for an EDLC, where it almost looks to be a triangle shape. It is noticeable that there is a vertical and sudden voltage drop before the discharging process. This is due to the presence of internal resistance in the EDLC. Apart from that, the linearity of the discharge slope, indirectly, tells us that this EDLC has a capacitive behavior. Unlike EDLC, the pseudocapacitor or battery will have a nonlinear discharge curve. The shape of 1000th curve is more triangle than the 15th and 1st cycles. This is normal because a supercapacitor needs a certain cycle number before it reaches some stability.

From the curve in Figure 11, various storage properties can be extracted where it is shown in Figure 12. Figure 12a shows the specific capacitance, *C_s_*, of the full SCOBY-based EDLC. The *C_s_* for the first is 7.44 F/g. The EDLC then experienced some specific capacitance loss about 12.6% after 50 complete charge–discharge cycles. Beyond the 50th cycle, the *C_s_* seemed to be stable up to 1000 cycles, with an average *C_s_* of (6.70 ± 0.10) F/g. The trend of *C_s_* is harmonized with the charge–discharge curve in Figure 10, where the consistency and stability are verified, as 250th cycle has almost identical curve with 1000th cycle. It is normal that a freshly fabricated EDLC has unbalanced performance within a few initial cycles before stability can be achieved. The trend of *C_s_* being slightly deviated from consistent value is due to the internal resistance. Table 2 shows the reported EDLC with various materials.

Figure 12b illustrates the *ESR* of the SCOBY-based EDLC obtained from 1000 charge–discharge cycles. *ESR* is an indicator of the internal resistance of the EDLC, which comes from the gap between the electrolyte and electrode themselves, current collector, external wires/clipper, and circuit. Electrode/electrolyte interaction also contributes to the *ESR* value. Another important parameter, which is responsible for the drop voltage, is the inclusion of PVDF to the EDLC electrode part, which is a good insulator. *ESR* for the first is 84.7 ohms. *ESR* of the EDLC seems to have fluctuated with the lowest and highest values at 72.7 and 100.5 ohm, respectively, where the pattern is less stable than the trend of *C_s_*. Free ions tend to develop into ion triplets/pairs/aggregates at a fast charge–discharge rate and high cycle number. These types of formations require a high current to transfer charges and can block other free ions from reaching the surface of the electrodes [46,47].

Another vital storage property of an EDLC is energy density. It reveals how much energy an EDLC can store per kilogram. Figure 12c reveals the *E_den_* of the full SCOBY-based EDLC throughout 1000 cycles. It is expected that the pattern of *E_den_* is similar to *C_s_*, where the first cycle is 1.03 Wh/kg, and it undergoes about a 12.4% reduction after 50 cycles. The *E_den_* then achieved stability at an average of 0.93 Wh/kg until 1000 cycles. This portrays that mobile ions in the EDLC experienced a consistent energy barrier after the 50th cycle. Unlike *E_den_*, *P_den_* is how much energy can be transferred per kilogram. *P_den_* for the constructed EDLC is shown in Figure 12d, where the first released is 176.5 W/kg. In Figure 12b, the *P_den_* trend is harmonized with the *ESR* trend owing to their relation in the charge transfer mechanism. By referring to Equation (8), *P_den_* is highly influenced by *ESR* and inversely proportional to it. The *P_den_* fluctuated between 148.9 W/kg and 206 W/kg.

#### 3.2.2. Storing and Energy Storage Mechanism

Cyclic voltammetry is another excellent tool to check capacitive behavior, electrochemical stability, and specific capacitance. CV analysis was performed for freshly fabricated EDLC after 444 charge–discharge cycles and after 1000 charge–discharge cycles in order to verify the pattern of the performance for charge–discharge. Figure 13 shows the CV plot of the EDLC at fast and slow scan rates. In all CV plots, it can be observed that fresh EDLC has a leaf-like shape CV plot. This tallies with the imperfect triangle-shaped charge–discharge curve for the first cycle in Figure 11. As the cycle number increased to 444 and 1000, the CV plot transformed into a rectangular-like shape. The rectangular-like shape CV plot indicates that the EDLC holds a good capacitive characteristic [48]. At initial cycles, most ions in the polymer matrix are in random distribution, thus, there is slower current response to voltage reversal at each end potential. This explains the tail on the top right part of the fresh EDLC CV plot. As the EDLC is charged and discharged for more cycles, proper charge-double layer is achieved leading to a rapid current response to voltage reversal, consequently altering the CV shape to a more rectangular shape.

Other than that, the EDLC is highly influenced by the scan rate. It is noticeable that the shape of the CV plot changed to a leaf-like shape when using a high scan rate. Ions can form a proper charge-double layer at a slower scan rate than a faster one, thus providing higher capacitance as well as rectangular-shaped CV. At a fast rate, the flow of ions toward the electrode is not stable. This causes some ions to recombine and form ion pairs/triplets. The formation of ion pair/triplets blocks the conduction of ions, which increases the internal resistance. This explains the deviation of CV plots at high scan rate. Table 3 reveals that the scan rates highly influence the specific capacitance.

## 4. Conclusions

In conclusion, it is safe to claim that the web-like network bacterial cellulose (BC) derived from SCOBY in this work is a multifunctional material. The BC presence contributed to both the film properties of the electrolytes and electrodes. A free-standing, pliable, and eco-friendly supercapacitor was successfully fabricated using polymer from bacterial cellulose and multiwalled carbon nanotubes as the active material for polarization. The optimum conductivity of the prepared BC-montmorillonite (MMT)-sodium bromide (NaBr) electrolyte system was (1.09 ± 0.02) × 10^−3^ S/cm with the assistance of 30 wt.% NaBr. From XRD analysis, the crystallinity of the electrolyte was reduced by blending BC and MMT and adding a proper amount of NaBr. As verified using FTIR analysis, the most complexation and interaction in the electrolyte system occurs in the hydroxyl region, where the peak position shift was significant. The best conducting SCOBY-based electrolyte breakdown and has ionic transference numbers at 1.48 V and 0.97, respectively. The constructed EDLC has excellent and stable storage properties throughout the 1000 cycles. The average specific capacitance and energy density of the EDLC were 6.70 F/g and 0.93 Wh/kg, respectively. The pattern of power density was highly related to the trend of the internal resistance of the EDLC. The CV plot of the EDLC showed no evidence of redox reaction. Apart from that, the performance of the EDLC is highly influenced by the scan rate.

## 5. Patents

N.A. Halim reports financial support was provided by Universiti Pertahanan Nasional Malaysia. N.A. Halim has the patent Flexible Conductive Composite Sheet and Method Thereof pending to Universiti Pertahanan Nasional Malaysia. N.A. Halim has the patent Flexible EDLC assembled from modified bacterial cellulose pending to Universiti Pertahanan Nasional Malaysia.

## Figures and Tables

**Figure 1 polymers-14-03196-f001:**
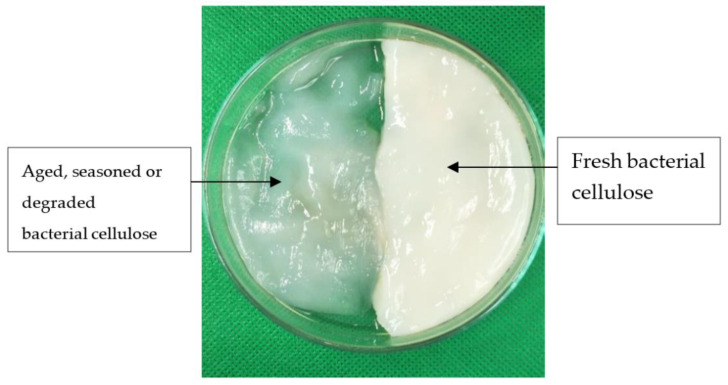
Fresh and degraded BC.

**Figure 2 polymers-14-03196-f002:**
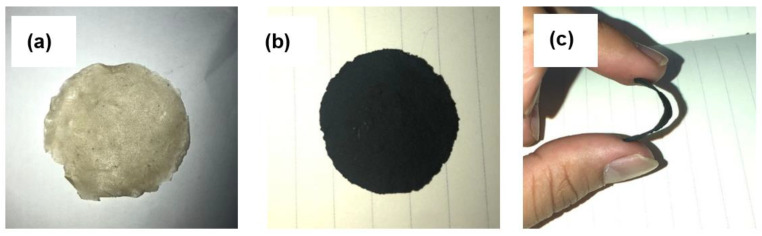
The formed (**a**) BXD electrolyte, (**b**) BXC electrode, and (**c**) bent BXC electrode.

**Figure 3 polymers-14-03196-f003:**
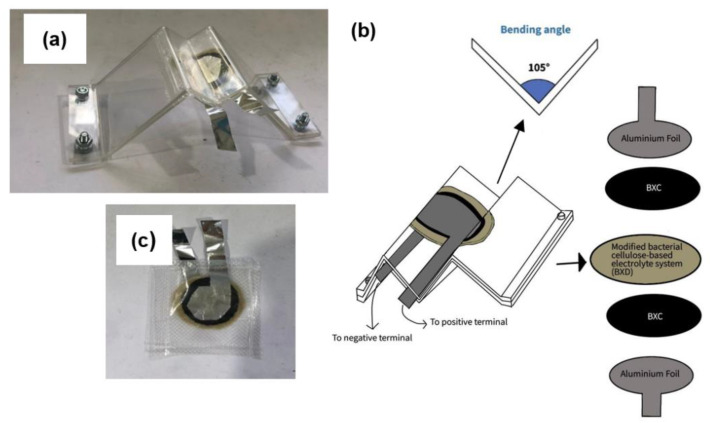
(**a**) Experimental setup for the full SCOBY-based EDLC, (**b**) schematic diagram of the full SCOBY-based EDLC, and (**c**) full SCOBY-based EDLC packed and sealed in a nonconductive plastic wrap.

**Figure 4 polymers-14-03196-f004:**
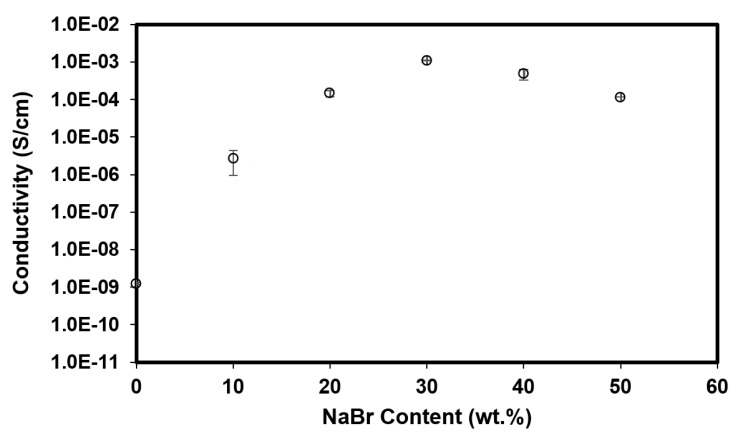
The effect of NaBr addition on the ionic conductivity of BC-MMT Films.

**Figure 5 polymers-14-03196-f005:**
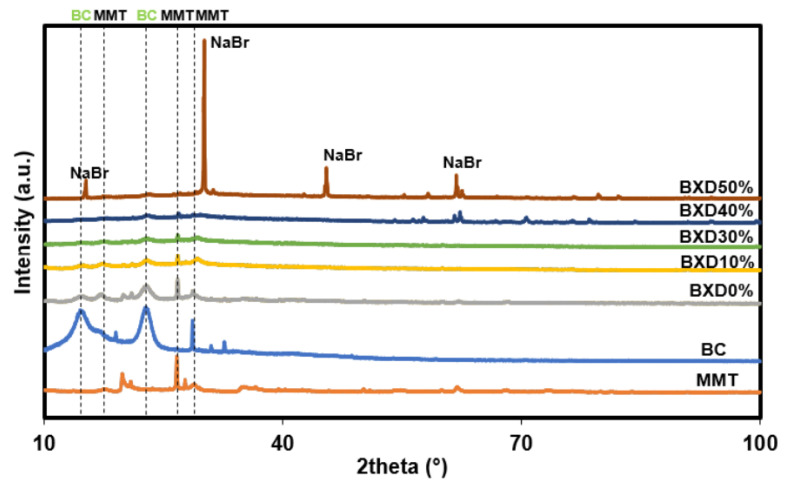
XRD spectrum of MMT, BC, and salted BC-MMT films.

**Figure 6 polymers-14-03196-f006:**
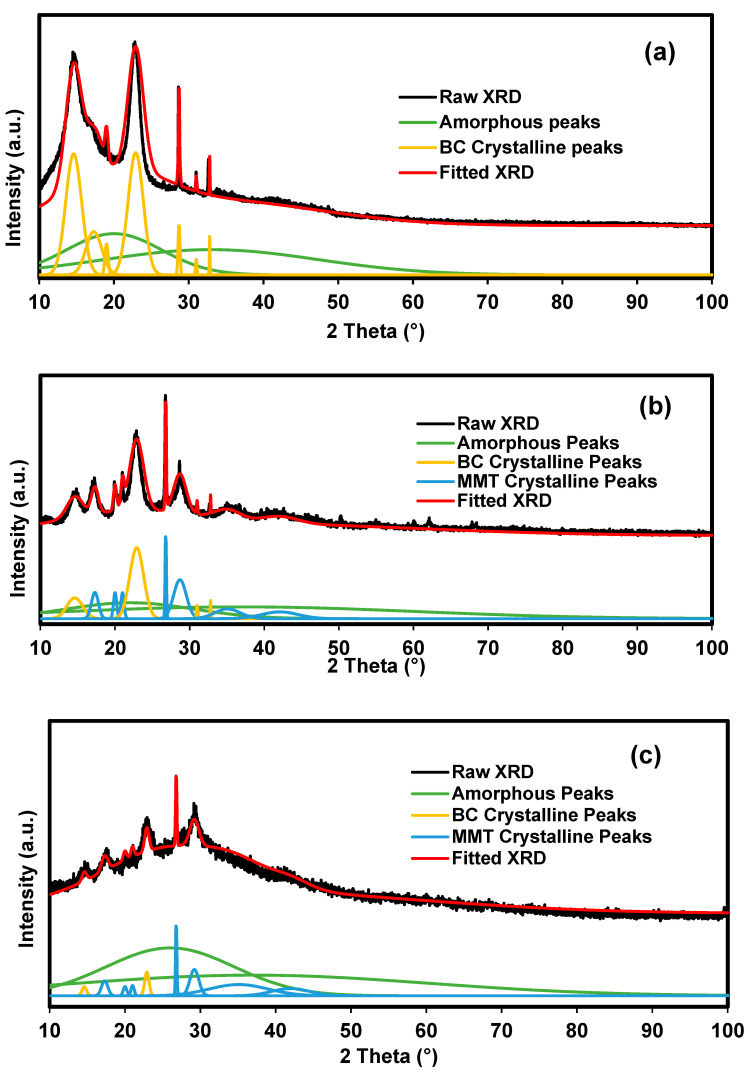
Deconvoluted XRD pattern of (**a**) BC, (**b**) BXD0%, and (**c**) BXD30%.

**Figure 7 polymers-14-03196-f007:**
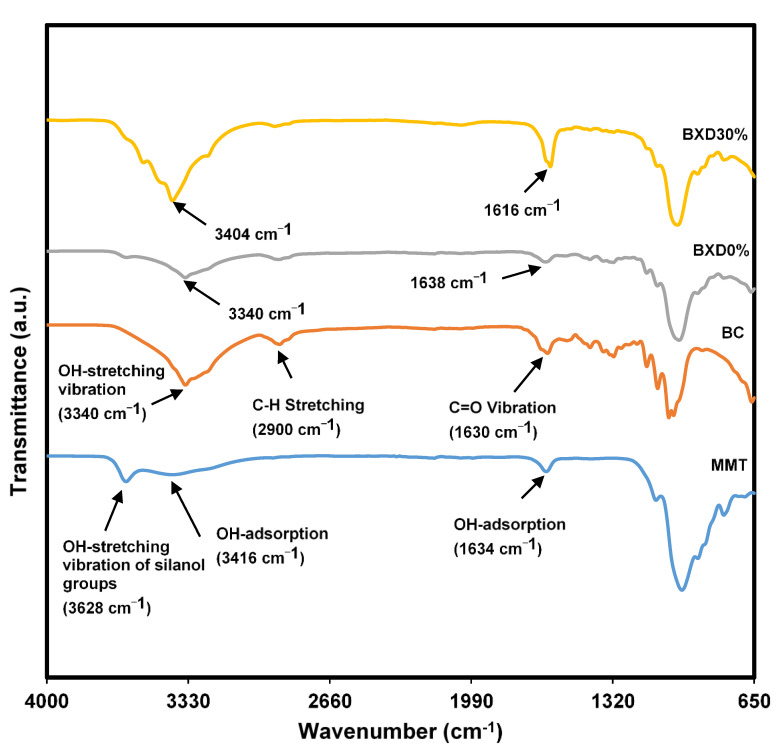
FTIR spectrum of MMT, BC, BXD0%, and BXD30%.

**Figure 8 polymers-14-03196-f008:**
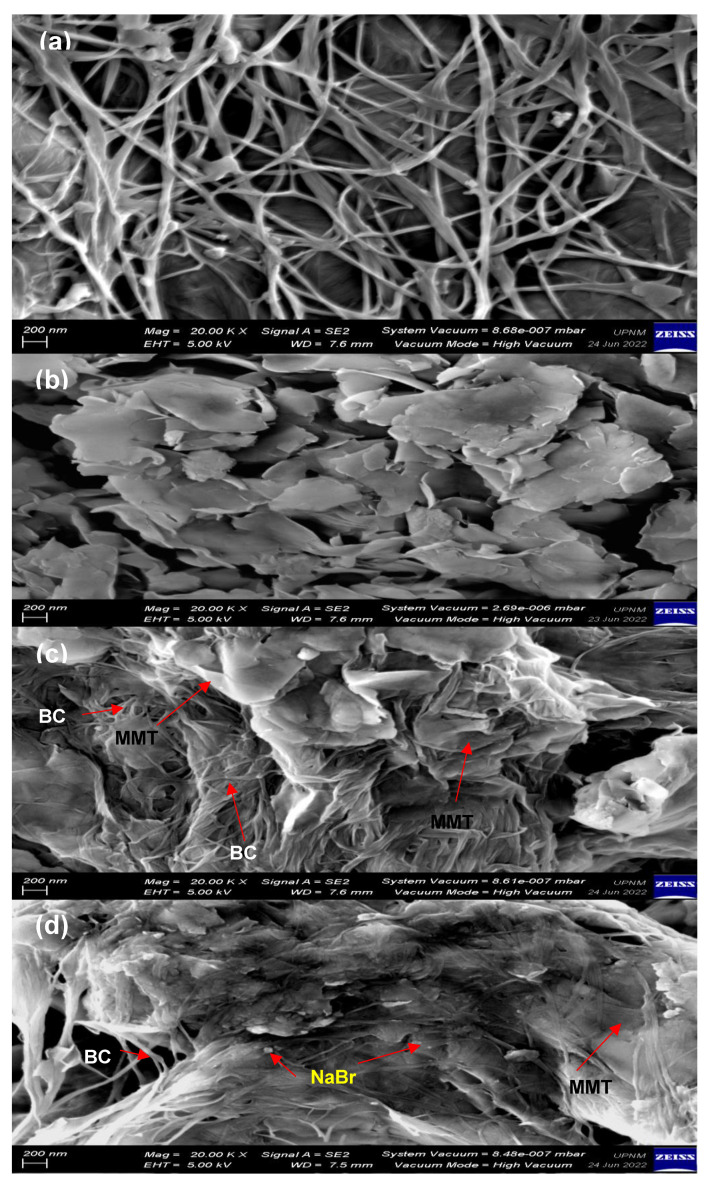
FESEM images of (**a**) BC, (**b**) MMT, (**c**) BXD0%, and (**d**) BXD30%.

**Figure 9 polymers-14-03196-f009:**
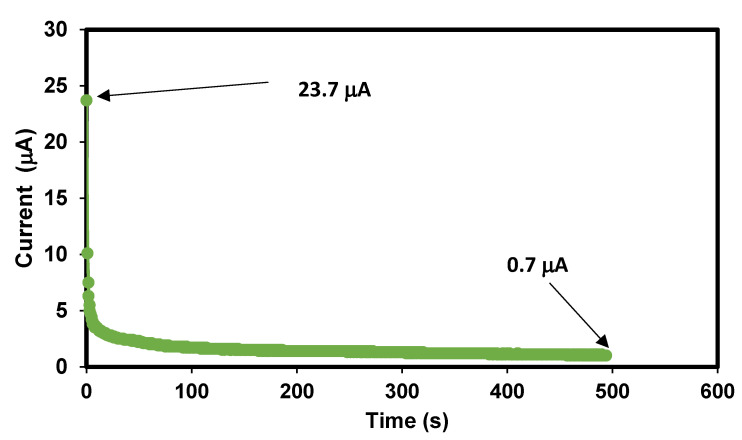
Polarization curve of BXD30% at a working voltage of 0.20 V.

**Figure 10 polymers-14-03196-f010:**
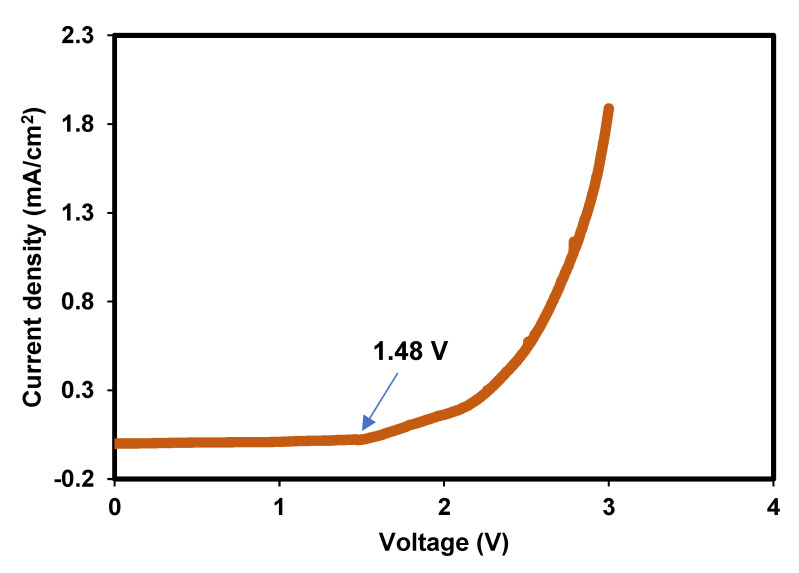
LSV plot of BXD30% at a sweeping rate of 10 mV/s.

**Figure 11 polymers-14-03196-f011:**
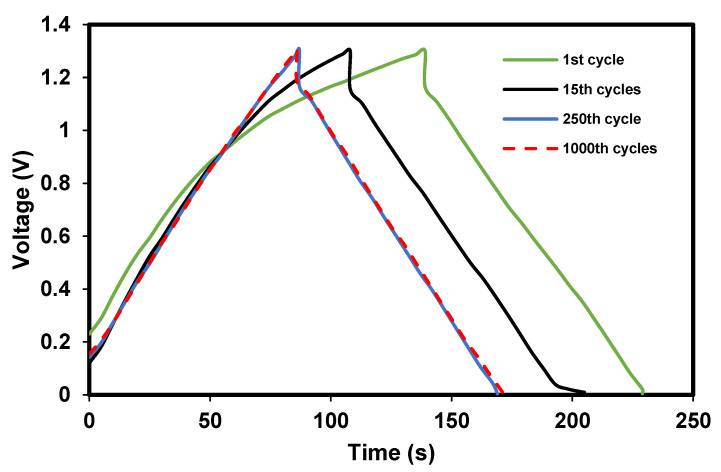
Charge–discharge plot for the fabricated supercapacitor at various cycle numbers.

**Figure 12 polymers-14-03196-f012:**
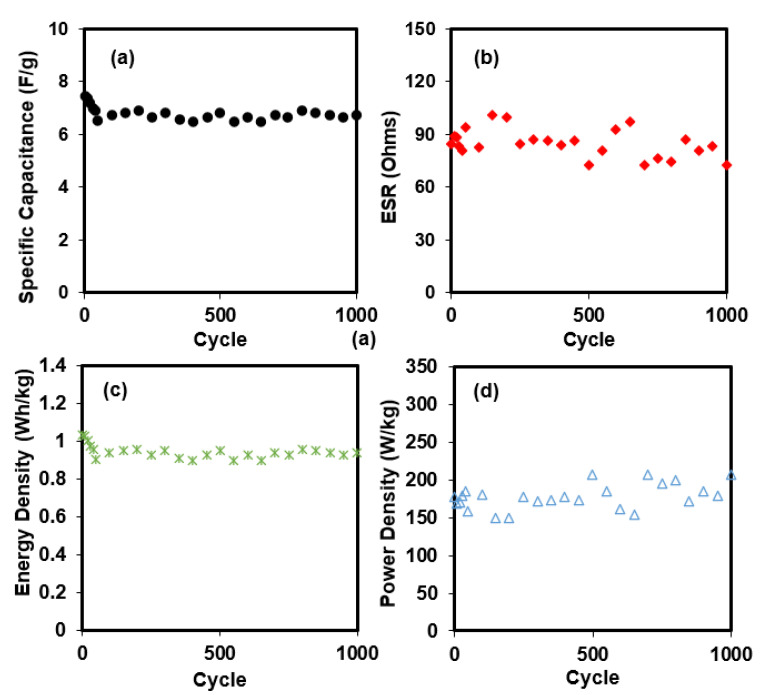
Crucial storage properties obtained from charge-discharge analysis such as (**a**) specific capacitance, (**b**) *ESR*, (**c**) energy density, and (**d**) power density.

**Figure 13 polymers-14-03196-f013:**
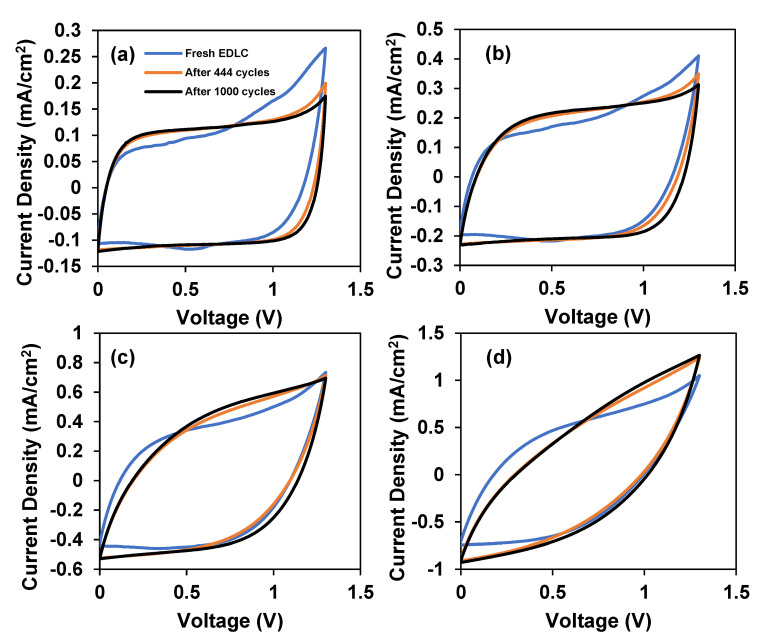
Cyclic voltammetry plot for the fabricated supercapacitor with a scan rate of (**a**) 10 mV/s, (**b**) 20 mV/s, (**c**) 50 mV/s, and (**d**) 100 mV/s.

**Table 1 polymers-14-03196-t001:** Degree of crystallinity of selected samples.

Sample	*χ_c_* (%)
BC	33.5
BXD0%	27.3
BXD30%	10.9

**Table 2 polymers-14-03196-t002:** Reported EDLCs with various polymer electrolytes and carbon electrodes.

Electrolyte	Electrodes	*C_s_* (F/g)	Cycles	Reference
PMMA-LiBOB	Carbon	0.52	50	[40]
Methylcellulose-NH_4_NO_3_	Activated Carbon	1.67	-	[41]
PEO-Mg(Tf)_2_ + EMITf	MWCNT-AB-PVdF-HFP	2.6–3.0	-	[42]
PVA–LiClO4	Activated carbon	3.0	200	[43]
Chitosan-PEO-NH_4_SCN	Carbon	3.8	-	[44]
PVA-Dextran-NH_4_I	Activated carbon-AB-PVdF	4.2	100	[45]
BC-MMT-NaBr	BC-MWCNT	6.7	1000	This work

**Table 3 polymers-14-03196-t003:** The specific capacitance after a certain cycle numbers.

	Specific Capacitance (*C_cyc_*)			
	10 mV/s	20 mV/s	50 mV/s	100 mV/s
Fresh EDLC	2.67	2.31	1.68	1.07
After 444 cycles	2.76	2.43	1.72	1.05
After 1000 cycles	2.78	2.75	1.86	1.14

## Data Availability

The data presented in this study are available on request from the corresponding author.

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
