# Peer review of "Multifunction Web-like Polymeric Network Bacterial Cellulose Derived from SCOBY as Both Electrodes and Electrolytes for Pliable and Low-Cost Supercapacitor"

_polymers, 2022, doi:10.3390/polym14153196_

Round 1
Reviewer 1 Report
This paper reported the systhesis of bacterial cellulose (BC)-based polymer for flexible and free standing supercapacitor applications. Although the structural and electrochemical properties of bacterial cellulose-based electrode and electrolyte were analyzed in detail, the work was not suitable for publication in this journal. The main reasons were as follows:
1. The paper was not well written and organized, and the usage of English language was rather unsatisfactory.
2. The work did not demonstate significant advance in the design of electrode, electrolyte materials and the applications of flexible advices. Compared to previous studies on flexible supercapacitors, there was no new insights into the electrochemical performance and mechanism of bacterial cellulose (BC)-based supercapacitors. Therefore, the novelty was very weak.
3. The equation (3) was wrong, the scan rate and the mass of active material were omitted. What does the "g" in equation (4) stand for?
4. There was no any results and discussion on Fig. 7.
5. Why did all the curves in Fig. 8 exhibit large Vdrop?
6. The electrochemical behavior tested under 1000th cycle did not make any sense, 5000th cycle was required at least.
7. The discussion on eletrochemical properties and storage mechanism were rather superficial. An more in-depth explanation should be provided.
8. The mechanical properties of flexible BC-based electrolyte and electrode should be given.
9. The electrochemical performance of the flexible BC-based supercapacitor under different deformation conditions should be discussed.
Reviewer 2 Report
1 In the manuscript, the authors provided naturally abundant polymer-based supercapacitors for energy storage applications. I do not suggest this manuscript for publication as it is. A major revision is required in the electrochemistry part.
1. The authors should take care of the notations used in the manuscript and all are in the same format. Especially subscript and superscript.
22. Page 9, Line 257 …This is due to the presence of internal resistance in the EDLC……, Explanation about the reason behind the voltage drop is not enough. What kind of internal resistance?? The author should clearly state this with references.
33. ……. The shape of the 1000th curve is more triangle than 15th and 1st cycle. This is normal because a supercapacitor needs a certain cycle number before it reaches some stability….. What about the performance/voltage drop of the EDLC after 1000th cycle.
44. The fabricated EDLC must run minimum 10,000 cycles to check the practical applicability.
55. The manuscript states that …..It is normal that a freshly fabricated EDLC has unbalanced performance within a few initial cycles before stability can be achieved….. But from Figure 9a, it is visible that the capacitance is not stable or has a gradual increase/decrease. Clarify this in the revised manuscript.
66. More clarification is required with the sentence …In few initial cycles, ions from the electrolyte and carbon electrodes are still in the process of recognizing the pattern polarization and electrostatic force among them….
77. Figure caption must be appropriate. (Figure 9, provide what is (a)-(d))
88. The comparison provided in Table 1 is inappropriate, the result should be compared with recent literature. Must replace with recent studies.
99. The energy density and the power density are very less compared to the recent carbon EDLCs. So, what is the novelty, or how are the authors going to sort out this problem
1. In all CV plots, it can be observed that fresh EDLC has leaf-like shape CV plot…… Authors should approach CV studies more scientifically. Kindly avoid this like comparison from the manuscript and point out scientifically the reason behind that.
111. It is not appropriate to conclude with a statement “stability of the device/EDLC”. Careful observation and explanation are required for the shift in the voltammogram with cycle number, do a post-mortem study after the cycles 444 and 1000, and explain the reason.
112. Table 2 is missing in the manuscript.
113. With the increase in scan rate, the shift in the voltammograms can be explained with ion/electron transportation rate, focus more on the reasons behind the shift, and include it in the manuscript.
Reviewer 3 Report
The presented manuscript discusses the possibility of using bacterial cellulose to create a low cost supercapacitor. CNT and MMT are used as additives. When reading the manuscript, the question arises about their distribution in the system. I would recommend the authors to provide electron microscopy data. The manuscript contains a number of typos and inaccuracies that need to be corrected. In my opinion, it is desirable to check the data of X-ray diffraction analysis again, for example, to use a larger amount of substance for scanning. I recommend removing "SCOBY" from the name, this term will not be clear to most readers. Authors should choose one format to designate of carbon nanotubes "CNT" or "CNTs". 2.1. materials For BC, more details must be provided, including the degree of polymerization. It is also important to indicate the method of removing bacteria and their residues, etc. Line 95. "local store" - this information is not informative for the reader, it is desirable to replace or delete it. Line 118. "ti" needs to be corrected. Line 133. It is necessary to remove the extra point. 3.1.1. Ionic Conductivity Analysis for The Bacterial Cellulose Electrolytes. In my opinion, starting a section with a picture is not entirely appropriate. It is better to move the picture below, after the first mention of the picture. For BC the angular positions of the basal reflexes (~14.6°, ~16.6° and ~22.7°) - https://doi.org/10.3390/pr8020171 . For diffraction patterns, you must specify the capture method. Lines 195, 196. "This is a proof of interaction between polymer chain with MMT via hydrogen bond." - I don't understand how the structure of BC strips was destroyed. What mechanism?
Reviewer 4 Report
The article is devoted to the study of the prospects for the use of a polymer based on bacterial cellulose (BC), obtained from a symbiotic culture of bacteria and yeast (SCOBY) as electrodes and electrolyte for the manufacture of a flexible and autonomous supercapacitor. In general, the presented study is quite interesting and promising not only from a fundamental point of view, but also from a further practical application. The article corresponds to the declared journal and can be accepted for publication in the future after the authors answer a number of questions that have arisen during its analysis.
1. The abstract needs to be improved, the authors should reflect in more detail the novelty and practical significance of the work.
2. The authors showed on a living example that the resulting material is quite flexible and plastic, but it is required to show numerically these parameters.
3. What is the reason for such a large difference in capacity at different NaBr content?
4. X-ray diffraction data require additional explanations and presentation of calculations, in particular, it is necessary to show the content of the observed phases, the degree of crystallinity and the amorphous phase.
5. The authors should explain what causes such sharp differences in the change in the current strength in Figure 6?
6. Conclusion requires significant revision and optimization, as well as reflection of further research prospects.
Round 2
Reviewer 1 Report
Some issues still remains unsettled. Since the bacterial cellulose (BC)-based polymer and yeast (SCOBY) are used as both electrodes and electrolytes to fabricate a flexible supercapacitor, its electrochemical performance under different deformation conditions should be evaluated. Also, the electrochemical behavior tested should be done at least 5000 cycles, instead of future work.
Reviewer 2 Report
The authors can improve the manuscript better.
Reviewer 3 Report
The authors of the manuscript answered the questions posed and took into account the comments. The editor can then evaluate the work for publication. Below are additional recommendations: Line 64. "EDLC" - you need to decipher the abbreviation. Line 183. It might be better to replace "empty polymer" with "cellulose" Figures 5, 6. I suggest that the authors show in the figures the area up to 70°. This will make the diffractograms more informative. Line 315. Delete extra point.
Reviewer 4 Report
The authors answered all questions, the article can be accepted for publication.
Round 3
Reviewer 1 Report
I have no further comments.
Reviewer 2 Report
The authors kindly remove the corrections from the manuscript and the manuscript is now considered for publication.